# NEURAL GROUNDPLANS: PERSISTENT NEURAL SCENE REPRESENTATIONS FROM A SINGLE IMAGE

**Prafull Sharma[1][*]**          **Ayush Tewari[1]**          **Yilun Du[1]**

**Sergey Zakharov[4]**          **Rares Ambrus[4]**          **Adrien Gaidon[4]**

**William T. Freeman[1,5]**   **Frédo Durand[1]**   **Joshua B. Tenenbaum[1,2,3]**   **Vincent Sitzmann[1]**

[1]MIT CSAIL   [2]MIT BCS   [3]MIT CBMM   [4]Toyota Research Institute
[5]The NSF AI Institute for Artificial Intelligence and Fundamental Interactions
`prafullsharma.net/neural_groundplans`

## ABSTRACT

We present a method to map 2D image observations of a scene to a persistent 3D scene representation, enabling novel view synthesis and disentangled representation of the movable and immovable components of the scene. Motivated by the bird's-eye-view (BEV) representation commonly used in vision and robotics, we propose *conditional neural groundplans*, ground-aligned 2D feature grids, as persistent and memory-efficient scene representations. Our method is trained self-supervised from unlabeled multi-view observations using differentiable rendering, and learns to complete geometry and appearance of occluded regions. In addition, we show that we can leverage multi-view videos at training time to learn to separately reconstruct static and movable components of the scene from a single image at test time. The ability to separately reconstruct movable objects enables a variety of downstream tasks using simple heuristics, such as extraction of object-centric 3D representations, novel view synthesis, instance-level segmentation, 3D bounding box prediction, and scene editing. This highlights the value of neural groundplans as a backbone for efficient 3D scene understanding models.

## 1 INTRODUCTION

We study the problem of inferring a persistent 3D scene representation given a few image observations, while disentangling static scene components from movable objects (referred to as dynamic). Recent works in differentiable rendering have made significant progress in the long-standing problem of 3D reconstruction from small sets of image observations (Yu et al., 2020; Sitzmann et al., 2019b; Sajjadi et al., 2021). Approaches based on pixel-aligned features (Yu et al., 2020; Trevithick & Yang, 2021; Henzler et al., 2021) have achieved plausible novel view synthesis of scenes composed of independent objects from single images. However, these methods do not produce persistent 3D scene representations that can be directly processed in 3D, for instance, via 3D convolutions. Instead, all processing has to be performed in image space. In contrast, some methods infer 3D voxel grids, enabling processing such as geometry and appearance completion via shift-equivariant 3D convolutions (Lal et al., 2021; Guo et al., 2022), which is however expensive both in terms of computation and memory. Meanwhile, bird's-eye-view (BEV) representations, 2D grids aligned with the ground plane of a scene, have been fruitfully deployed as state representations for navigation, layout generation, and future frame prediction (Saha et al., 2022; Philion & Fidler, 2020; Roddick et al., 2019; Jeong et al., 2022; Mani et al., 2020). While they compress the height axis and are thus not a full 3D representation, 2D convolutions on top of BEVs retain shift-equivariance in the ground plane and are, in contrast to image-space convolutions, free of perspective camera distortions.

Inspired by BEV representations, we propose *conditional neural groundplans*, 2D grids of learned features aligned with the ground plane of a 3D scene, as a persistent 3D scene representation

---
[*]Email: prafull@mit.edu, sitzmann@mit.edu

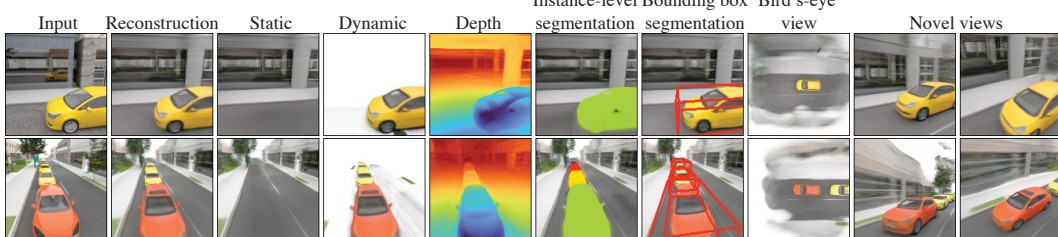

Figure 1: Given a single image, our model infers separate 3D representations for static and dynamic scene elements, enabling high-quality novel view synthesis with plausible completion, unsupervised instance-level segmentation, 3D bounding box prediction, 3D scene editing, and extraction of object-centric 3D representations. Our model is trained self-supervised using unlabeled multi-view videos.

reconstructed in a feed-forward manner. Neural groundplans are a hybrid discrete-continuous 3D neural scene representation (Chan et al., 2022; Peng et al., 2020; Philion & Fidler, 2020; Roddick et al., 2019; Mani et al., 2020) and enable 3D queries by projecting a 3D point onto the groundplan, retrieving the respective feature, and decoding it via an MLP into a full 3D scene. This enables self-supervised training via differentiable volume rendering. By compactifying 3D space with a nonlinear mapping, neural groundplans can encode unbounded 3D scenes in a bounded region. We further propose to reconstruct separate neural groundplans for 3D regions of a scene that are movable and 3D regions of a scene that are static given a single input image. This requires that objects are moving in the training data, enabling us to learn a prior to predict which parts of a scene are movable and static from a single image at test time. We achieve this additional factorization by training on multi-view videos, such as those available from cameras at traffic intersections or sports game footage. Our model is trained self-supervised via neural rendering without pseudo-ground truth, bounding boxes, or any instance labels. We demonstrate that separate reconstruction of movable objects enables instance-level segmentation, recovery of 3D object-centric representations, and 3D bounding box prediction via a simple heuristic leveraging that connected regions of 3D space that move together belong to the same object. This further enables intuitive 3D editing of the scene.

Since neural groundplans are 2D grids of features without perspective camera distortion, shift-equivariant processing using inexpensive 2D CNNs effectively completes occluded regions. Our model thus outperforms prior pixel-aligned approaches in the synthesis of novel views that observe 3D regions that are occluded in the input view. We further show that by leveraging motion cues at training time, our method outperforms prior work on the self-supervised discovery of 3D objects.

In summary, our contributions are:

- We introduce self-supervised training of conditional neural groundplans, a hybrid discrete-continuous 3D neural scene representation that can be reconstructed from a single image, enabling efficient processing of scene appearance and geometry directly in 3D.
- We leverage object motion as a cue for disentangling static background and movable foreground objects given only a single input image.
- Using the 3D geometry encoded in the dynamic groundplan, we demonstrate single-image 3D instance segmentation and 3D bounding box prediction, as well as 3D scene editing.

## 2 RELATED WORK

**Neural Scene Representation and Rendering**. Several works have explored learning neural scene representations for downstream tasks in 3D. Emerging neural scene representations enable reconstruction of geometry and appearance from images as well as high-quality novel view synthesis via differentiable rendering. A large part of recent work focuses on the case of reconstructing a *single* 3D scene given dense observations (Cheng et al., 2018; Tung et al., 2019; Sitzmann et al., 2019a; Lombardi et al., 2019; Mildenhall et al., 2020; Yariv et al., 2020; Tewari et al., 2021). Alternatively, differentiable rendering may be used to supervise encoders to reconstruct scenes from a single or few images in a feedforward manner. Pixel-aligned conditioning enables reconstruction of compositional scenes (Yu et al., 2020; Trevithick & Yang, 2021), but does not infer a compact 3D representation. Methods with a single latent code per scene do, but do not generalize to compositional scenes (Sitzmann et al., 2019c; Jang & Agapito, 2021; Niemeyer et al., 2020; Sitzmann et al., 2021; Kosiorek

et al., 2021). Voxel grid based approaches offer both benefits, but are computationally costly(Lal et al., 2021; Sajjadi et al., 2021; Dupont et al., 2020). Hybrid discrete-continuous neural scene representations offer a compromise by factorizing a dense 3D field into several lower-dimensional representations that are used to condition an MLP (Chan et al., 2022; Chen et al., 2022a). In particular, neural groundplans and axis-aligned 2D grids enable high-quality unconditional generation of 3D scenes (DeVries et al., 2021; Chan et al., 2022) as well as reconstruction of 3D geometry from pointclouds (Peng et al., 2020). We similarly use axis-aligned 2D grids of features for self-supervised scene representation via neural rendering, but reconstruct them directly from few or a single 2D image observations.

**Bird's-Eye View Representations**. Bird's-eye view has been explored as a 3D representation in vision and robotics, particularly for autonomous driving applications. Prior work uses ground-plane 2D grids as representations for object detection and segmentation (Saha et al., 2022; Harley et al., 2022; Philion & Fidler, 2020; Reiher et al., 2020; Roddick et al., 2019), layout generation and completion (Cao & de Charette, 2022; Jeong et al., 2022; Mani et al., 2020; Yang et al., 2021b), and next-frame prediction (Hu et al., 2021; Zięba et al., 2020). The bird's-eye view is generated either directly without 3D inductive biases (Mani et al., 2020), or similar to our proposed approach, by using 3D geometry-driven inductive biases such as unprojection into a volume (Harley et al., 2022; Chen et al., 2022b; Roddick et al., 2019), or by generating a 3D point cloud (Philion & Fidler, 2020; Hu et al., 2021). However, prior approaches are supervised, using ground truth bounding boxes or semantic segmentation as supervision. In contrast, we present a self-supervised conditional groundplan representation, learned only from images via neural rendering. While we show that our self-supervised representation can be used for rich inference tasks using simple heuristics, our method may be extended for more challenging tasks using the techniques developed in prior work.

**Dynamic-Static Disentanglement**. Our work is related to prior work on learning to disentangle dynamic objects and static background. Some prior work leverages object motion across video frames to learn separate representations for movable foreground and static background in 2D (Kasten et al., 2021; Ye et al., 2022; Bao et al., 2022), while other recent work can also learn 3D representations (Yuan et al., 2021; Tschernezki et al., 2021). Our approach is similar in using object motion as cue for disentanglement and multi-view as cue for 3D reconstruction, but uses it as supervision to train an encoder-based approach that enables reconstruction from a *single* image instead of scene-specific disentanglement from multiple video frames.

**Object-centric Scene Representations**. Prior work has aimed to infer object-centric representations directly from images, with objects either represented as localized object-centric patches (Lin et al., 2020; Eslami et al., 2016; Crawford & Pineau, 2019; Kosiorek et al., 2018; Jiang et al., 2019) or scene mixture components (Engelcke et al., 2020; Burgess et al., 2019; Greff et al., 2019; 2016; 2017; Du et al., 2021a), with the slot attention module (Locatello et al., 2020) increasingly driving object-centric inference. Resulting object representations may be decoded into object-centric 3D representations and composed for novel view synthesis (Yu et al., 2022; Smith et al., 2022; Elich et al., 2022; Chen et al., 2021; Bear et al., 2020; Zakharov et al., 2020; 2021; Beker et al., 2020; Du et al., 2021b). BlockGAN and GIRAFFE (Nguyen-Phuoc et al., 2020; Niemeyer & Geiger, 2021) build unconditional generative models for compositions of 3D-structured representations, but are restricted to only generation. Some methods rely on annotations such as bounding boxes, object classes, 3D object models, or instance segmentation to recover object-centric neural radiance fields (Ost et al., 2021; Yang et al., 2021a; Guo et al., 2020; Yang et al., 2022). Several scene reconstruction methods (Zakharov et al., 2020; 2021; Beker et al., 2020; Nie et al., 2020) use direct supervision to train an object representation and detector to infer an editable 3D scene from a single frame observation. Kipf et al. (2021) leverage motion as a cue for self-supervised object disentanglement, but do not reconstruct 3D and require additional conditioning in the form of bounding boxes. In this work, we demonstrate that a representation factorized into static and movable 3D regions can serve as a powerful backbone for object discovery. While not explored in this work, slot attention and related object-centric algorithms could be run on our already sparse groundplan of movable 3D regions, faced with a dramatically easier task than when run on images directly.

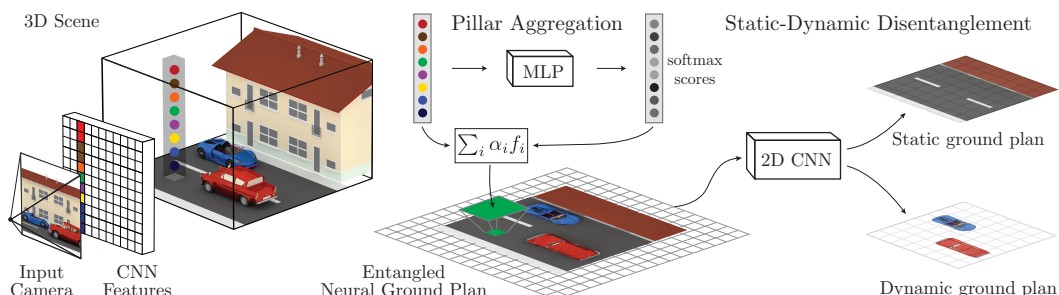

Figure 2: **Groundplan inference.** Given a context image, we first extract a set of CNN features. We unproject the features into 3D and re-sample them at "pillars" on top of the location of groundplan vertices. Pillars are aggregated into groundplan features using a softmax-weighted sum. The resulting 2D grid of features is decomposed into separate dynamic and static groundplans by a 2D CNN. The coordinate-encoding MLP is not visualized in this figure. Please refer to Sec. 3 for details.

## 3   CONDITIONAL NEURAL GROUNDPLANS

In this section, we describe the process of inferring a neural ground plan given one or several image observations in a feed-forward manner, as well as subsequent novel view synthesis. The method will be trained on a dataset of multi-view videos with calibrated cameras with wide baselines. Please see Fig. 2 for an overview.

**Compactified neural groundplans for unbounded scene representations**. A neural groundplan is a 2D grid of features aligned with the ground plane of the 3D scene, which we define to be the $xz-$plane. A 3D point is decoded by projecting it onto the groundplan and retrieving the corresponding feature vector using bilinear interpolation. This feature is then concatenated with the vertical $y$-coordinate of the query point and decoded into radiance and density values via a fully connected network, enabling novel view synthesis using volume rendering (Mildenhall et al., 2020). In this definition, however, it is only possible to decode 3D points that lie within the boundaries of the neural groundplan, which precludes reconstruction and representation of unbounded scenes. We thus compactify $\mathbb{R}^3$ by implementing a non-linear coordinate re-mapping as proposed by Barron et al. (2021). Points $\mathbf{x}$ within a radius $r_{\text{inner}}$ around the groundplan origin remain unaffected, but points outside this radius are contracted. For any 3D point $\mathbf{x}$, the contracted 3D coordinate can be computed as $\mathbf{x}' = C(\mathbf{x}) = ((1+k) - k/||\mathbf{u}||)(\mathbf{u}/||\mathbf{u}||)r_{\text{inner}}$, where $\mathbf{u} = \mathbf{x}/r_{\text{inner}}$, and $k$ is a hyperparameter which controls the size of the contracted region. Note that $C$ is invertible, such that $\mathbf{x} = C^{-1}(\mathbf{x}')$ is a function that takes a 3D point in contracted space $\mathbf{x}'$ to the original 3D point $\mathbf{x}$ in linear space.

**Reconstructing neural groundplans from images**. Inferring a neural groundplan from one or several images proceeds in three steps: (1) feature extraction, (2) feature unprojection, (3) pillar aggregation. Given a single image $\mathbf{I}$, we first extract per-pixel features via a CNN encoder to yield a feature tensor $\mathbf{F}$. We define the camera as the world origin and center the neural groundplan accordingly, approximately aligned with the ground level. The image features are unprojected to a 3D feature volume $\mathbf{v}$ in contracted world space using the inverse of the contract function defined earlier. We extract the feature at a contracted 3D point $\mathbf{x}'$ in $\mathbf{v}$ as $\mathbf{v}(\mathbf{x}') = \mathbf{F}[\pi(C^{-1}(\mathbf{x}'))]$, where $C^{-1}(\mathbf{x}')$ first maps the contracted point to linear world space and $\pi(\cdot)$ projects it onto the image plane of the context view using camera extrinsics and intrinsics. At any vertex of the groundplan, the discretized $y$-coordinates of the volume form a "pillar". Next, we aggregate each pillar into a point to create the 2D groundplan. We first use a coordinate-encoding MLP $D(\cdot)$ to transform the volume as $\mathbf{f}(\mathbf{x}') = D(\mathbf{v}(\mathbf{x}'), \mathbf{x_c}, \mathbf{d})$, where $\mathbf{x_c}$ denotes the 3D point in linear camera coordinates of the context camera, and $\mathbf{d}$ denotes the ray direction from the camera center to that point. Since all features along a camera ray are identical in $\mathbf{v}$, coordinate encoding is used to add the depth information to the features. In the case of multi-view input images, the volumes corresponding to each input view are mean pooled. Associated to each 2D vertex of the groundplan is now a set of features $\{\mathbf{f}_i\}_{i=1}^N$, where $N$ is the number of samples along the $y$-dimension. We use a "pillar-aggregation" MLP to compute softmax scores as $\alpha_i = P(\mathbf{f}_i, \mathbf{x}_i)$, where $P(\cdot)$ denotes the MLP and $\mathbf{x}_i$ is the linear coordinate of the $i$-th point on the pillar. Finally, the features are aggregated by computing the weighted sum of the features, $\mathbf{g} = \sum_i \alpha_i \mathbf{f}_i$.

**Differentiable Rendering**. We can render images from novel camera views via differentiable volume rendering (DeVries et al., 2021; Lombardi et al., 2019; Mildenhall et al., 2020). To resolve points closer to the camera more finely, we adopt logarithmic sampling of points along the ray with more

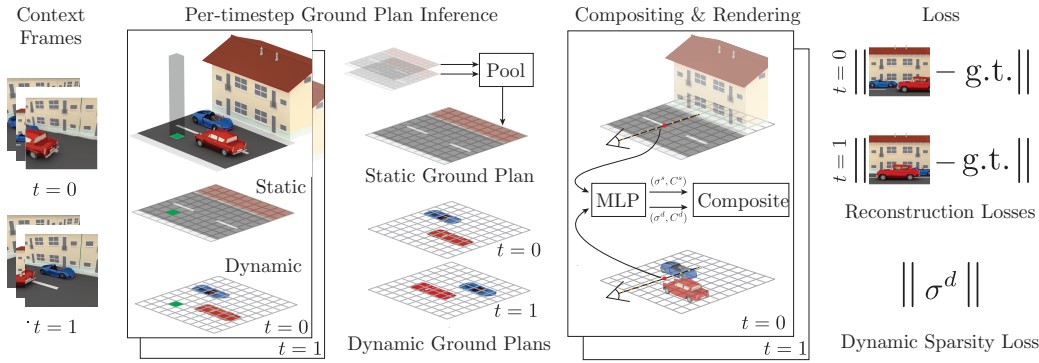

Figure 3: **Learning Static-Dynamic Disentanglement.** Given multiple frames of a video, we extract per-frame, compactified, static and dynamic groundplans according to Fig. 2. Static groundplans are pooled into a time-invariant groundplan. We then composite per-frame dynamic and static time-invariant groundplans via differentiable volume rendering. Our model is supervised only via a re-rendering loss on video frames. We encourage the model to explain as much of the scene density as possible with the static groundplan via a sparsity loss on per-frame dynamic volume rendering densities. The surface loss is not visualized here.

samples close to the camera (Neff et al., 2021). For each sampled point $\mathbf{x}$ on the camera ray, we need to compute its density and color for volume rendering. This is accomplished using a rendering MLP, as $(\mathbf{c_x}, \sigma_\mathbf{x}) = R(\mathbf{g_x}, y_\mathbf{x})$, where $R(\cdot)$ denotes the MLP, $\mathbf{g_x}$ are the groundplan features for the point $\mathbf{x}$ computed by projecting the query 3D coordinates onto the groundplane and bilinearly interpolating the nearest grid points, and $y_\mathbf{x}$ is the y value of the sampled point $\mathbf{x}$.

# 4 LEARNING STATIC-DYNAMIC DISENTANGLEMENT

We now describe training on multi-view video to learn to disentangle static and dynamic components of the scene. Furthermore, we describe a method for performing self-supervised 3D object discovery and 3D bounding box prediction using the geometry encoded in the dynamic groundplan representation. Please see Fig. 3 for an overview of the multi-frame training for static-dynamic disentanglement.

**Disentangling static and dynamic neural groundplans**. We leverage the fact that objects move in the given multi-view videos as the training signal. We pick two random frames of a video. For each frame, we infer an *entangled* neural groundplan as described in the previous section. Features in this entangled neural groundplan parameterize both static and dynamic features of the scene, for instance, a car as well as the road below it. We feed this groundplan into a fully convolutional 2D network, which disentangles it into two separate groundplans containing *static* and *dynamic* features. The per-frame static groundplans are mean-pooled to obtain a single, time-independent static groundplan.

**Compositing groundplans**. To render a scene using the disentangled static and dynamic groundplans, we first decode query points using both groundplans, yielding *two* sets of (density, color) values for each point. We use the compositing operation proposed by Yuan et al. (2021) to compose the contribution from static and dynamic components along the ray. Given the color and density for static $(\mathbf{c}^S, \sigma^S)$ and dynamic $(\mathbf{c}^D, \sigma^D)$ parts, the density of the combined scene is calculated as $\sigma^S + \sigma^D$. The color at the sampled point is computed as a weighted linear combination $w^S \mathbf{c}^S + w^D \mathbf{c}^D$, where $w^S = (1 - \exp(-\delta\sigma^S))/(1 - \exp(-\delta(\sigma^S + \sigma^D)))$, $w^D = (1 - \exp(-\delta\sigma^D))/(1 - \exp(-\delta(\sigma^S + \sigma^D)))$, and $\delta$ is the distance between adjacent samples on the camera ray.

**Losses and Training**. We train our model on multi-view video, where multi-view information is used to learn the 3D structure, while motion is used to disentangle the static and dynamic components in the scene. During training, we sample two time-steps per video. For each time-step, we sample multiple images from different camera views; some of the views are used as input to the method while others are used to compute the loss function. We use the input images to infer static and dynamic groundplans, and use them to render out per-frame query views. Our per-frame loss consists of an image reconstruction term, a hard surface constraint, and a sparsity term.

$$\mathcal{L} = \underbrace{||R - I||_2^2 + \lambda_{\text{LPIPS}}\mathcal{L}_{\text{LPIPS}}(R, I)}_{\mathcal{L}_{\text{img}}} - \underbrace{\lambda_{\text{surface}} \sum_i \log(\mathbb{P}(w_i))}_{\mathcal{L}_{\text{surface}}} + \underbrace{\lambda_{\text{sparse}} \sum_i |\sigma_i^D|}_{\mathcal{L}_{\text{dyn\_sparsity}}}. \quad (1)$$

| | CLEVR (1 view) | | | CoSY (1 view) | | | CoSY (5 views) | | |
|---|---|---|---|---|---|---|---|---|---|
| | PSNR↑ | SSIM↑ | LPIPS↓ | PSNR↑ | SSIM↑ | LPIPS↓ | PSNR↑ | SSIM↑ | LPIPS↓ |
| Ours | **34.5** | **0.956** | **0.15** | **15.71** | **0.43** | **0.53** | **18.29** | **0.57** | **0.43** |
| PixelNeRF | 33.98 | 0.945 | 0.200 | 14.61 | 0.34 | 0.64 | 17.31 | 0.49 | 0.50 |
| uORF | 29.35 | 0.898 | 0.151 | — | — | — | — | — | — |

Table 1: Quantitative baseline comparison of novel view synthesis results. We outperform Pixel-NeRF (Yu et al., 2020) and uORF (Yu et al., 2022) in terms of PSNR, SSIM, and LPIPS on both CLEVR and CoSY datasets.

$\mathcal{L}_{\text{img}}$ measures the difference between the rendered and ground truth images, $\mathbf{R}$ and $\mathbf{I}$ respectively, using a combination of $\ell_2$ and patch-based LPIPS perceptual loss. $\mathcal{L}_{\text{surface}}$ encourages both static and dynamic weight values (the weight for each sample in the rendering equation) $w_i$ for all samples along the rendered rays to be either $0$ or $1$, encouraging hard surfaces (Rebain et al., 2022). Here, $\mathbb{P}(w_i) = \exp(-|w_i|) + \exp(-|1 - w_i|)$. The sparsity term $\mathcal{L}_{\text{dyn\_sparsity}}$ takes as input densities decoded from the dynamic groundplan for all the rendered rays, and encourages the values to be sparse. This forces the model to explain most of the non-empty 3D structure as possible via the static groundplan and only expressing the moving objects using the dynamic groundplan, leading to reliable static-dynamic disentanglement. Without this loss, the model could explain the entire scene with just the dynamic component. The loss functions are weighed using the hyperparameters $\lambda_{\text{LPIPS}}$, $\lambda_{\text{surface}}$, and $\lambda_{\text{sparse}}$. While we describe the loss functions for a single sample of ground-truth and rendered image, in practice, we construct mini-batches by randomly choosing multiple views of a scene at different time steps, and evaluate the loss function on each sample.

**Unsupervised object detection and extracting object-centric 3D representations**. Our formulation yields a model that maps a single image to two radiance fields, parameterizing static and dynamic 3D regions respectively. Please see Fig. 1 for an example. We now perform a search for connected components in the dynamic neural groundplan to perform 3D instance-level segmentation, monocular 3D bounding box prediction, and the extraction of object-centric 3D representations. Specifically, given a dynamic groundplan, we first sample points in a 3D grid around the groundplan origin and decode their densities. We now perform conventional connected-component labeling in the groundplan space using accumulated density values, identifying the disconnected dynamic objects. We perform 2D instance-level segmentation for a queried viewpoint using volume rendering based on the densities expressed by the dynamic groundplan and assigning a color to the points corresponding to each of the identified objects, see Fig. 1 for an example. Furthermore, we compute the smallest box that contains the connected component to get a 3D bounding box for each identified object. Finally, we crop tiles of the dynamic groundplan that belong to a given object instance to obtain object-centric 3D representations, enabling editing of 3D scenes such as deletion, insertion, and rigid-body transformation of objects. This approach is not limited to a fixed number of objects during training or at test time. As we will show, this simple method is at par with the state of the art on self-supervised learning of object-centric 3D representations, uORF (Yu et al., 2022). Note that our approach is compatible with prior work leveraging slot attention (Locatello et al., 2020; Kipf et al., 2021) and other inference modules, which can be run on the disentangled dynamic groundplan which, in contrast to image space, enables shift-equivariant processing free from perspective distortion and encodes 3D structure. For implementation details, refer to Appendix A.

## 5 RESULTS

We demonstrate that our method infers a 3D scene representation from a single image while disentangling static and dynamic components of the scene into static and dynamic groundplans respectively. We then show that connected components analysis suffices to leverage the densities in the dynamic groundplan for instance-level segmentation, bounding box prediction, and scene editing.

**Datasets**. Our method is trained on multi-view observations of dynamic scenes. We present results on the moving CLEVR dataset (Yu et al., 2022), commonly used for self-supervised object discovery benchmarks (Yu et al., 2022), and the procedurally generated autonomous driving dataset CoSY (Bhandari, 2018). CoSY enables generation of a high-quality, path-traced dataset of multi-view videos with large camera baselines. We rendered multi-view observations of 9000 scenes with moving cars, sampled using 15 background city models and 95 car models. We train on 8000 scenes, and

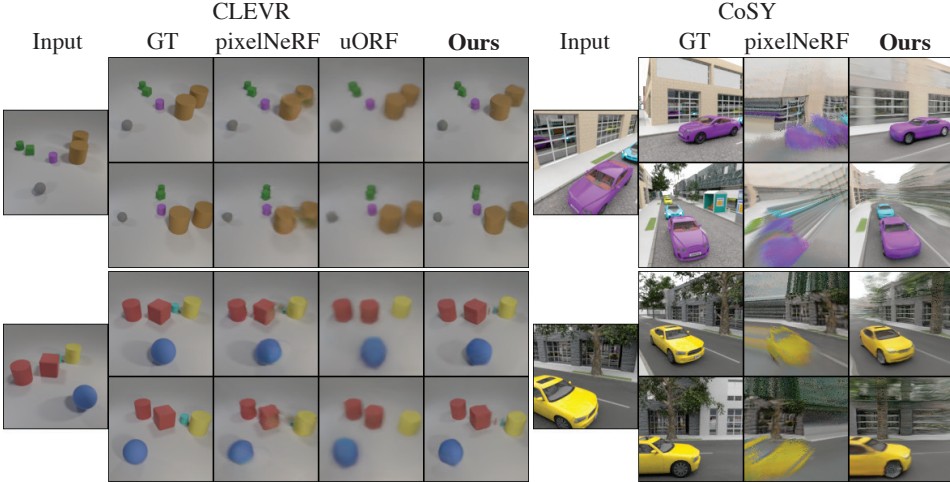

Figure 4: **Qualitative comparisons.** Comparison for novel-view synthesis given a single context view with PixelNeRF (Yu et al., 2020) and uORF (Yu et al., 2022).

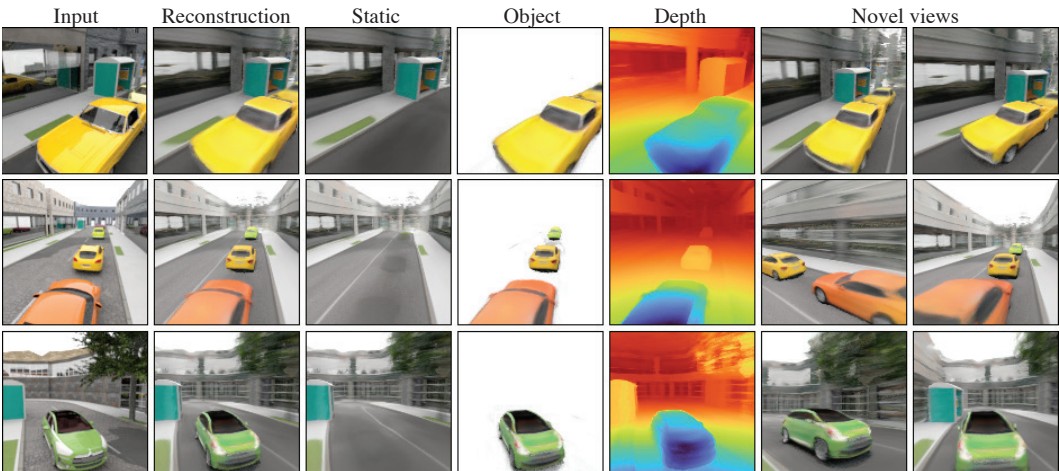

Figure 5: **Single-image reconstruction, disentanglement of static and dynamic objects, and novel view synthesis.** Given a single input image, our method can disentangle the observed scene into static and object components based on what the model observed as not-moving and moving in the training data. In these examples, the cars are isolated in the object component as the model was training on video data of cars moving on the road.

evenly split the rest into validation and test sets. Further details about dataset generation are presented in the Appendix A.6. Datasets and code will be made publicly available.

**Novel View Synthesis and Scene Completion**. We present novel views rendered from groundplans inferred from a single image from CLEVR and CoSY. For single-shot 3D reconstruction and novel view synthesis, we compare against PixelNeRF (Yu et al., 2020), a state-of-the-art single-image 3D reconstruction method, and uORF (Yu et al., 2022), state-of-the-art unsupervised object-centric 3D reconstruction method. We train PixelNeRF models on our datasets using publicly available code. We finetune the uORF model pretrained on CLEVR on our CLEVR renderings, and train it from scratch on CoSY using publicly available code. Fig. 4 provides a qualitative comparison to PixelNeRF and uORF in terms of single-image 3D novel view synthesis on both CoSY and CLEVR. Note that our model produces novel views with plausible completions of parts of the scene that are unobserved in the context image such as the back-side of objects. As expected from a non-generative method, regions that are entirely unconstrained such as occluded parts of the background (such as buildings) are blurry. While PixelNeRF succeeds in novel view synthesis on CLEVR, renderings on the complex CoSY dataset show significant artifacts, possibly caused by the linear sampling employed by PixelNeRF. uORF does not synthesize realistic images when trained on CoSY. Please refer to the supplemental webpage for results of uORF on the CoSY (Appendix B.5). On CLEVR, uORF generally produces high-quality renderings, but lacks high-frequency detail. In contrast to

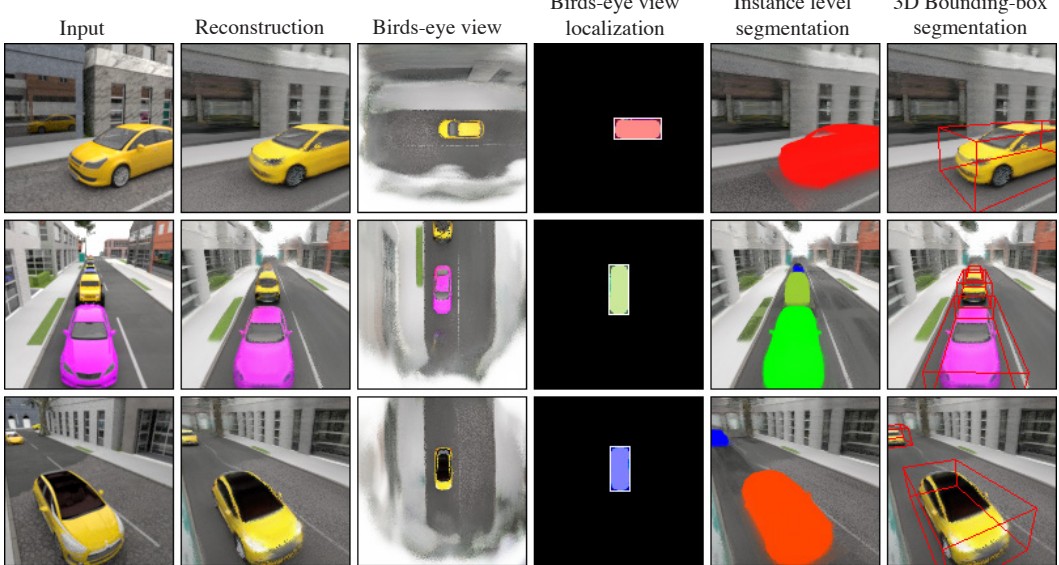

Figure 6: **Object Localization.** Given a single input image, we can use our inferred 3D representation of the scene to compute (a) bird's-eye view rendering, (b) object localization in the bird's-eye view space, (c) instance-level segmentation, and (d) 3D bounding box prediction.

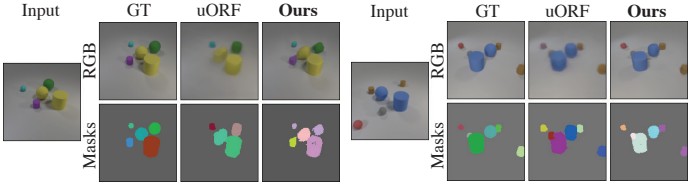

Figure 7: **Qualitative Comparison for instance-level segmentation.** Qualitative comparison of instance-level segmentation result with uORF (Yu et al., 2022).

|  | ARI↑ | NV-ARI↑ |
|---|---|---|
| Ours | **0.84** | **0.84** |
| uORF | 0.83 | 0.82 |

Table 2: Quantitative segmentation accuracy evaluation on CLEVR using ARI and NV-ARI metrics.

these methods, our method reliably synthesizes novel views with high-frequency detail for both datasets. Quantitatively, we outperform both methods on novel-view synthesis in terms of PSNR, SSIM, and LPIPS metrics on both datasets (refer to Table 1). Note that qualitatively, the performance gap to baseline methods is significantly larger than quantitative results would suggest. This is due to the fact that much of the pixels used to compute PSNR are observing scene regions far outside the frustum of the input view. Here, all methods fail to reconstruct the true 3D appearance and geometry, as it is completely uncertain given the context view, resulting in low PSNR numbers for all methods (refer to Appendix B.1). As can be seen in the qualitative results, our method achieves significantly better reconstruction quality in parts of the 3D scene that lie in the frustum of the input camera, even if these areas are occluded in the input view. Our method further succeeds at fusing information across multiple context views, increasing the quality of the renderings with an increasing number of context views from varied viewpoints (refer to Appendix B.2).

**Static-Dynamic Disentanglement**. Given only a single image, our method computes separate static and dynamic groundplans that can be used to individually render the static and movable parts of the scene respectively. Fig. 5 shows results on single-image reconstruction of static and movable scene elements. Note that cars are reliably encoded by the dynamic groundplan, and our method inpaints regions occluded in the input view.

**Instance-level Segmentation and Bounding Box prediction**. The separate reconstruction of movable scene components in the dynamic groundplan enables object detection via instance-level segmentation and bounding box prediction. Fig. 6 presents the instance-level segmentation and 3D bounding box prediction results of the proposed 3D object discovery via connected component discovery using the density inferred using the dynamic groundplan from the bird's-eye view. Fig. 7 provides a qualitative comparison of object discovery with uORF on CLEVR dataset. While uORF succeeds at segmenting CLEVR scenes with fidelity comparable to ours, it fails to provide reconstruction and instance-level segmentation for our diverse and visually complex street-scale CoSY dataset.

| Input | Reconstruction | Individual Objects | | Deletion | Addition | Rearrangement |

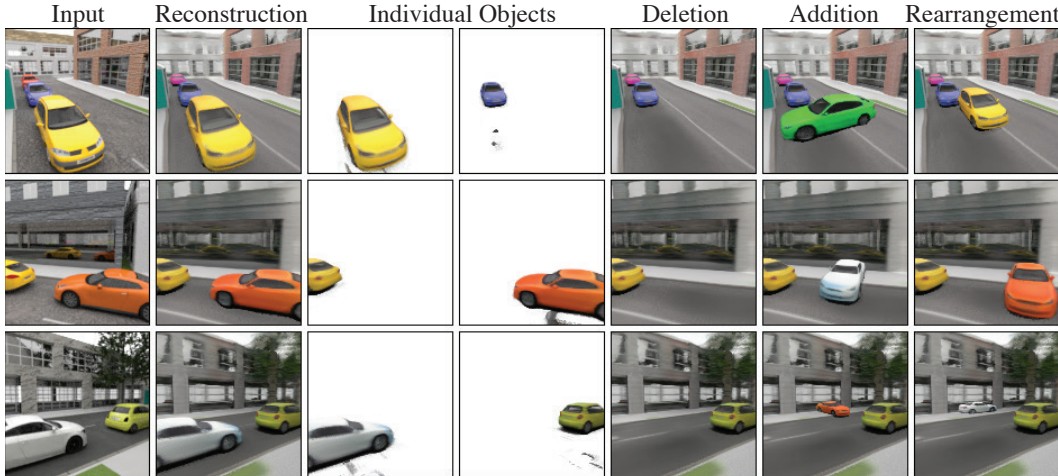

Figure 8: **Object-centric representations and scene editing.** The proposed static-dynamic neural groundplans enables object discovery using connected components labeling on the density of objects encoded in the dynamic groundplan (left). This enables straight-forward scene editing such as deletion, addition, or rigid-body transformation via directly editing the neural groundplan (right).

Our method reliably segments separate car instances and predicts the 3D bounding boxes, including for cars that are only partially observed. Table 2 quantitatively compares the computed segmentation maps on CLEVR to uORF. We use the Adjusted Rand Index (ARI) metrics following uORF. We evaluate this metric in the input view (ARI), as well as in a novel view (NV-ARI). We perform at par with uORF on both of these metrics, demonstrating that our 3D ground plan representation reaches state of the art results with simple heuristics. Please refer to the supplemental webpage for video results (Appendix B.5). In addition, as mentioned before, we achieve higher-quality novel-view synthesis results, and also achieve significantly better results on the challenging CoSY dataset. Since uORF is based on slot attention, it can only attend to a finite number of objects, whereas groundplans can support any number of objects and require a single forward pass to render all objects.

**Scene Editing**. Instance-level segmentation, dynamic-static disentanglement, and 3D bounding boxes enable straight-forward 3D editing, such as translation, rotation, deletion, and insertion of individual objects in the scene. Objects can be rotated by arbitrary angles by simple bilinear interpolation of the groundplan features (refer to Appendix B.3). As the dynamic groundplan does not encode static scene regions such as the street below cars, cars can easily be moved from one scene to another. Fig. 8 provides scene editing results of our method. Note that such editing is difficult with methods that lack a persistent 3D representation, such as PixelNeRF.

## 6  DISCUSSION

**Limitations and Future Work**. Although our method achieves high-quality novel view synthesis from a single image, generated views are not photorealistic, and unobserved scene parts are blurry commensurate with the amount of uncertainty. Future work may explore plausible hallucinations of unobserved scene parts. Future work may further explore the use of more sophisticated downstream processing of the groundplan to enable, for instance, prior-based inference of object-centric representations (Locatello et al., 2020). Finally, we plan to investigate the combination of the proposed approach with flow-based dynamics reasoning, which would negate the need for multi-view video.

**Conclusion**. Our paper demonstrates self-supervised learning of 3D scene representations that are disentangled into movable and immovable scene elements. By leveraging multi-view video at training time, we can reconstruct disentangled 3D representations from a single image observation at test time. We show the potential of neural ground plans as a representation that may enable data-efficient solutions to downstream processing of the 3D scene, such as completion, instance-level segmentation, 3D bounding box prediction, and 3D scene editing. We hope that our paper will inspire future work on the use of self-supervised neural scene representations for general scene understanding tasks.

## 7    ACKNOWLEDGEMENTS AND DISCLOSURE OF FUNDING

This work is in part supported by DARPA under CW3031624 (Transfer, Augmentation and Automatic Learning with Less Labels) and the Machine Common Sense program, Singapore DSTA under DST00OECI20300823 (New Representations for Vision), NSF CCRI # 2120095, NSF RI #2211259 the National Science Foundation under Cooperative Agreement PHY-2019786 (The NSF AI Institute for Artificial Intelligence and Fundamental Interactions, http://iaifi.org/), Stanford Institute for Human-Centered Artificial Intelligence (HAI), Stanford Center for Integrated Facility Engineering (CIFE), Qualcomm Innovation Fellowship (QIF), Samsung, Ford, Amazon, and Meta. The Toyota Research Institute also partially supported this work. This article solely reflects the opinions and conclusions of its authors and not other entity.

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

## A    IMPLEMENTATION DETAILS

In this section, we provide more training details of our method to enable reproducibility.

### A.1    NETWORK ARCHITECTURES

**CNN Encoder**. Following PixelNeRF, we use the first 4 convolutional blocks of the ResNet34 architecture to create a feature grid. For a $128 \times 128$ resolution image, this results in a feature grid of $64 \times 64$ resolution with $512$ features per point. We further use a shallow CNN to reduce the number of feature channels, and to aggregate information along the y-axis. This step helps us reduce the memory footprint for further processing in 3D. This feature aggregation CNN has two convolutional hidden layers with 256 hidden units and a stride of 2 along the height of the feature tensor. The output is a $16 \times 64$ resolution feature grid with $128$ features per point. We use ReLU as the activation function for all layers in the encoder.

**Unprojection**. We populate a coarse 3D feature volume of shape $64 \times 16 \times 64$ with 128-dimensional latents for points in the camera frustum by bilinearly sampling the aggregated encoder feature tensor. This volume is processed as explained in the main paper: a coordinate encoding MLP transforms each feature into a new 128 dimensional feature vector. This MLP has two hidden layers with 128 hidden units each and a ReLU activation after each hidden layer.

**Multi-view input**. In case of multi-view inputs, we average the unprojected latents for all 3D points in the canonical 3D coordinate system for all views. This results, like in the monocular case, in a 3D volume of $128 \times 64 \times 16 \times 64$ ($CH \times X \times Y \times Z$).

**Aggregation along the height**. To construct an entangled groundplan, we aggregate the 3D feature volume into a 2D groundplan using a pillar aggregation MLP. Consider a pillar of features orthogonal to the groundplan for a particular point $(x, z)$, in our case that is $128 \times 16$. The MLP takes in each latent with its corresponding 3D coordinate in the world coordinate system. It outputs the softmax scores for each of these latents which can be used to sum these latents along the pillar into a single 128-dimensional latent, as explained in the main paper.

**Disentanglement CNN**. We use a shallow CNN with 4 hidden convolutional layers to disentangle the entangled groundplan of shape $128 \times 64 \times 64$. The first two convolutional layers have 128 hidden units with kernel size of 3, stride of 1, and reflection padding of 1.

The last two convolutional layers comprise of 256 hidden units with the same configuration for other parameters. These are alsso followed by a $2\times$ bilinear upsampling layers. This shallow CNN outputs groundplan of shape $256 \times 256 \times 256$, in which the first 128 channels of the feature tensor are attributed as the static groundplan and the rest of the 128 channels are used as the dynamic groundplan for that timestep.

**Projections**. Our method performs project of a 3D point on the image plane to get the corresponding feature tensor. This is performed using the intrinsic matrix of the camera. For a given 3D point in camera coordinates $\mathbf{x_c}$ and intrinsic matrix $\mathbf{K}$, the resulting 2D point on the image plane is computed as $\mathbf{Kx}$. For projecting a 3D query point $\mathbf{x}$ on the groundplan, we use grid sample using $(x, z)$-coordinates of the 3D query points $\mathbf{x}$.

**Neural renderer**. Similar to PixelNeRF, we use a MLP as a renderer with 4 hidden layers that have 128 hidden units. Our renderer and the input latent are significantly smaller than the ones used in PixelNeRF, making our rendering cheaper.

We use two rendering MLPs, for coarse and fine sampling with 256 samples for the coarse MLP and 128 samples for the fine rendering along with 32 samples at the predicted depth based on the coarse renderings.

**Initialization**. We use the kaiming normal initialization for initializing all the weights. All biases were initialized with uniform distribution in $[-1e^{-3}, 1e^{-3}]$.

## A.2 HYPERPARAMETERS

We use Adam Kingma & Ba (2014) with a learning rate of $3e^{-4}$ to train our pipeline with the image reconstruction loss (L2), hard surfaces loss, and the alpha sparsity loss for 200 epochs. The losses were weighted by $\lambda_{\text{img}} = 1$, $\lambda_{\text{HSL}} = 0.1$, and $\lambda_{\text{sparse}} = 0.01$. The model was then further finetuned by adding the LPIPS loss weighted by $\lambda_{\text{lpips}} = 0.5$. We found that using LPIPS loss from the beginning of the training process made the training unstable. We sampled $1e^4$ rays to compute the loss for each training sample in the input batch in the initial phase. During the first phase of training, the rays are sampled randomly and in the second phase when LPIPS loss is applied to the training, we sample rays to render image patches of $16 \times 16$. LPIPS loss using VGG is applied to these patches by normalizing the range of the output RGB values to be between $[-1, 1]$. The model was trained on a single 32G V100 GPU with a batch of 4 input samples with 2 timesteps for N views (N=5).

## A.3 CURRICULUM TRAINING

Our model supports multi-view, as well as monocular reconstruction. We start our training only using multi-view reconstruction with 5 input views for the first 200 epochs, and then switch to a variable mode where the model is given a varying number of input views ranging between 1-5 views. This

curriculum approach helps the network to first learn the relevant scene priors, before learning to complete the 3D structure of the scene.

### A.4 HEURISTICS FOR LOCALIZATION

For localization of objects, we consider the dynamic groundplan and render it from an orthographic bird's-eye view. Along with the bird's-eye view RGB image, we also generate the occupancy map (per-pixel accumulated alpha) as shown in the manuscript in Fig. 5. Since the groundplans have a spatial extent of $256 \times 256$, the pphic occupancy map is rendered at the same resolution. The hard-surfaces loss encourages densities to be close to either 0 or 1. Thus, we threshold the density map with a threshold of 0.9. We find the regions of the map with connected components in the thresholded density map using `label` and `regionsprop` functions from the `sklearn` Pedregosa et al. (2011). To remove any remaining artifacts, we only keep the regions which have an area larger than 6 in the pixel space of the groundplan. This is a hyperparameter based on the size of the objects in the scene. Given the localization in the orthographic bird's-eye view, we can find the height of the objects by computing the depth within each region. This information gives us a 3D bounding box around each of the localized object. The instance level segmentation is produced via volume rendering, by overriding the RGB values within the detected region to a chosen color for the object. These predictions can be made more accurate, for example, by (1) further tuning these hyperparameters, (2) sampling the orthographic density map at a higher resolution, (3) increasing the resolution of the groundplans at training time, and (4) training with more samples per camera ray.

### A.5 SCENE EDITING

Scene editing is performed by editing the dynamic groundplan. Once the different objects have been localized in the dynamic groundplan, we can edit the dynamic groundplan to carry out object deletion, insertion, and rearrangement. Localizing an object gives us its features in the spatial region of the groundplan used for rendering. To delete the object, we replace the features in this regions with features from the dynamic groundplan that encode zero density. For inserting an object at a given location, we find the corresponding $(x, z)$ location in the dynamic groundplan and set the features at that location to the features corresponding to the object. Rearrangement of objects can be seen as a combination of deletion and insertion where we first delete the object from the existing location, followed by inserting the object at the new location in a possibly new orientation. To perform rotation of an object, we rotate the patch of features that correspond to the object to be rotated.

### A.6 DATASETS

**CLEVR**. We generate scenes using the default configuration from (Johnson et al., 2017) (BSD License[1]), using the rubber material and default object sizes. We render the scene with 6 different cameras, all at the same fixed distance from the origin as the camera used in CLEVR, but with azimuth angles increasing at 60 degree increments. Objects in the CLEVR dataset are captured at 2 timesteps, and are simulated to move a distance of 0.25 to 0.75 meters between the two timesteps. Images were rendered across 6 different cameras, with resolution of $128 \times 128$ using CYCLES renderer with 512 samples per pixel. Our dataset consists of 1500 samples, divided into 1000 train and 500 test samples.

**CoSY**. We develop this dataset using the city generation code provided by Bhandari (2018). We generate 15 different configurations of the city using CityEngine, with variations in building shapes, heights, and materials. This city layout is further processed in Blender using Python to add cameras, trees, bus stops, and moving cars. Cameras are sampled on hemispheres of radii in range [4-6m] a maximum height of 4m viewing the center of the circular base of the hemisphere located at randomly chosen points. We sample 15 cameras for each of the randomly chosen centers, with the varying radius of the hemisphere for each camera. The field-of-view for all sampled cameras is 60 degrees, with a symmetric sensor size of 32mm, resulting in a focal length of 110.85 in pixel space. Note that we never sample any bird's-eye view as the maximum height of the camera is clipped at 4m. We choose 50-100 cars to be spawned in different locations of the generated city. We use a fixed environment map with diffused white light. In addition to the rendering capabilities of CoSY, we add

---

[1] https://github.com/facebookresearch/clevr-dataset-gen/blob/main/LICENSE

Context
images                                                Views around the objects

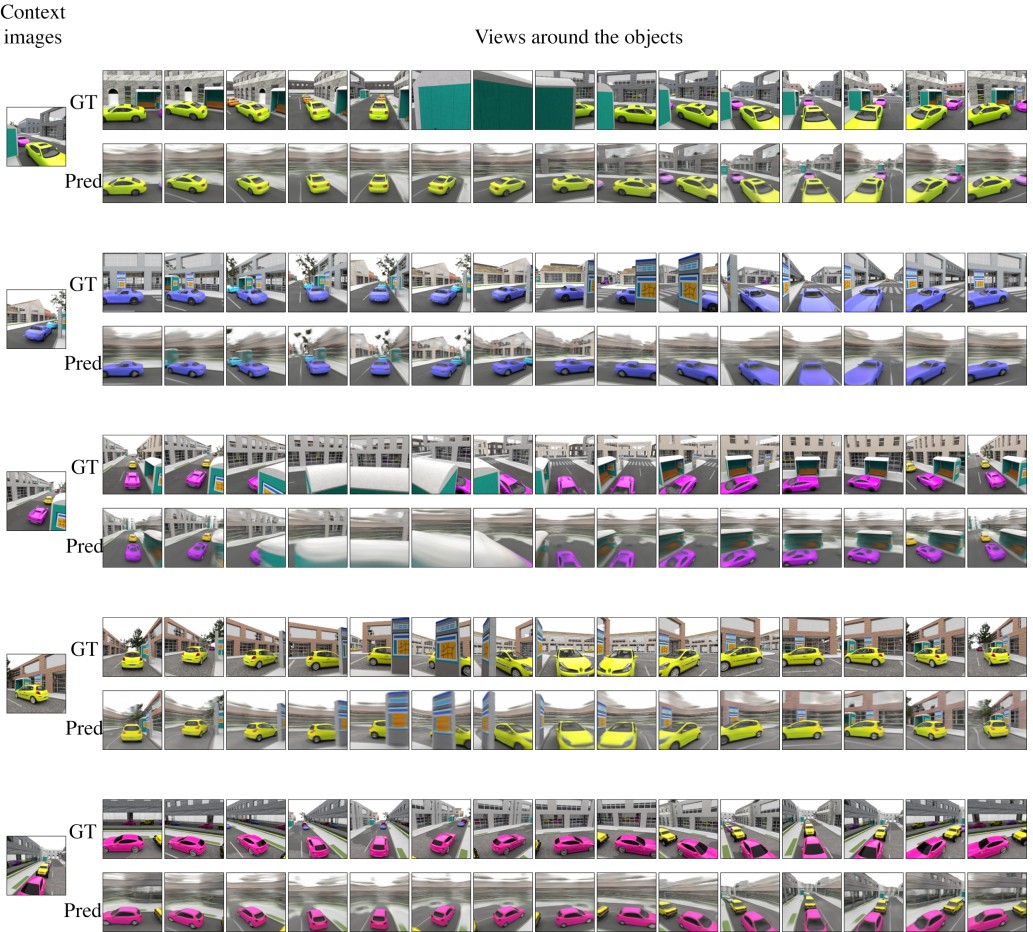

Figure 9: **Visual quality of views for a given input image.**

the ability to move cars over timesteps. Given the location and direction of the movement (direction where the car is facing), we change the location of the car over 10 timesteps to a sampled total translation from range 2-4 m. For each sampled camera, we render the 10 frames with a resolution of $128 \times 128$ using CYCLES renderer with 512 samples per pixel. Our dataset has 9000 such samples, which are further divided into 8000 train, 500 validation, and 400 test samples. Thus each sample has images, poses, focal length, and principal point for 15 cameras. The dataset will be publicly released for further research in this direction.

# B  ADDITIONAL RESULTS

## B.1  VISUALIZATION OF VIEWS OUTSIDE THE OBSERVED VIEW

Fig. 9 presents the output renderings for various target camera viewpoints for the given input image. We observe that as the target view shifts away from the context view, our method renders the geometry with appropriate texture of the car in the view, but outputs a blurry background as expected from a non-generative model.

## B.2  UNCERTAINTY WITH RESPECT TO THE NUMBER OF INPUT VIEWS

Fig. 10 provides results of street-scale scenes reconstructed using an increasing number of input images from different camera viewpoints. Our method successfully integrates information across observations into a single, multi-view consistent representation. The background reconstructions improve with more input views, as a result of less uncertainty for the occluded regions.

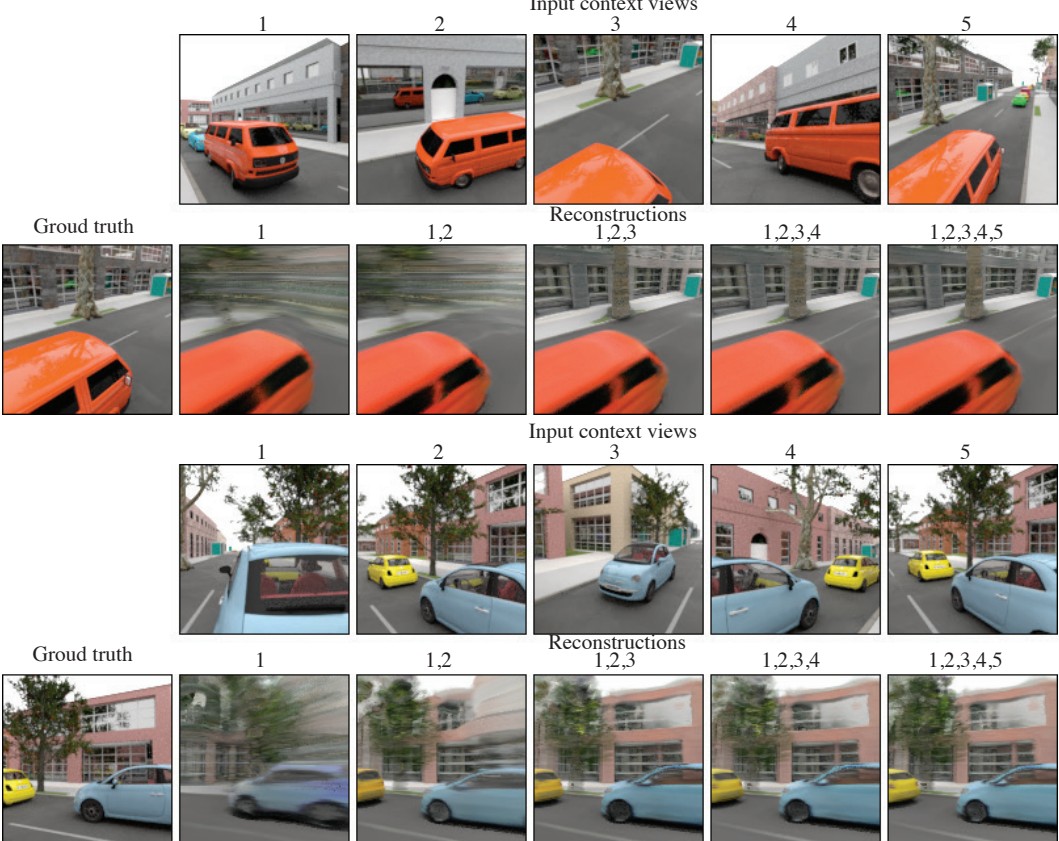

Figure 10: **Resolving uncertainty with increasing number of context views** Our model can take a variable number of viewpoints of a scene as input. We find that the model gets a good prior on the object from a single view but struggles to reconstruct high-quality details in the unobserved regions of the background. This uncertainty gets resolved with more input views.

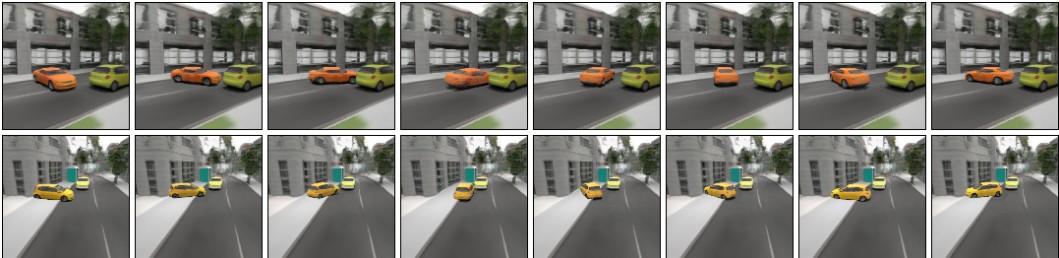

Figure 11: **Scene editing with rotated objects.** We present images of scenes with edited groundplans where cars are rotated at various angles in the ground plane.

Apart from effect of uncertainty in the unobserved regions on the visual quality, there are two factors that contribute to blurriness in the results. First, our method is in the regime of prior-based reconstruction from a few images. This is in contrast to the regime of single-scene overfitting (e.g. NeRF) in which prior work has demonstrated photo-realistic results. In our task setup of prior based reconstruction, a certain degree of uncertainty exists. For instance, there is uncertainty about the exact depth and geometry of the 3D scene given the input images. In these cases, the model will learn to blur proportionate to the amount of uncertainty. We note that this is not a limitation of our method specifically - all prior-based 3D reconstruction methods share this property. We outperform pixelNeRF, a strong baseline in this regime of 3D reconstruction from few images, both quantitatively and qualitatively. Second, our rendering quality is limited by computational cost. In contrast to single-scene methods, our method needs to fit not only the differentiable rendering in GPU memory, but also the whole inference pipeline - CNNs, 3D lifting, groundplans, etc. This limits the resolution of the groundplans (a car is expressed by $\sim 6$ latents) as well as the number of volume rendering samples. Increasing computational budget would lead to better renderings.

## B.3 ROTATING OBJECTS

Fig. 11 provides rendered images from edited groundplans where the cars are rotated at different angles. Note that the representation only allows for rotations in the $xz$-plane.

## B.4 MOTION INTERPOLATION

The dynamic groundplan can be further used to perform motion interpolation in 3D, using optical flow for groundplan interpolation prediction using off-the-shelf, state-of-the-art optical flow and frame interpolation methods. To demonstrate the efficacy of the groundplans for frame interpolation, we trained RIFE Huang et al. (2020) on our model trained on the GQN-rooms Eslami et al. (2018) dataset. We generated simple linear motion trajectories for objects over 10 frames. We added tall static cylinders as pillars in the room which generate occlusions. The scene was rendered from 15 different camera views. We first trained our model on the GQN dataset and used the output dynamic groundplans to train the frame interpolation method. The training was done in 2 steps. Firstly, we extracted dynamic groundplan for the samples over different timesteps by passing the multi-view observation through our groundplan generation pipeline. Two dynamic groundplans at different timesteps were given as input to RIFE, and an intermediate timestep was queried. An L2 loss on the output dynamic plan against the output of our pipeline for that timestep was sufficient to obtain a good initialization. In the training stage, we combined the RIFE model with our method for higher-quality results. All losses discussed in the main paper were applied on the rendered novel views generated using the output dynamic floorplans on the intermediate timesteps. In Fig. 12, we show the rendered output of our model for the intermediate timesteps given the leftmost and rightmost frames as input. The proposed method succeeds at inferring the correct object motion, and enables novel view synthesis through space and time.

## B.5 SUPPLEMENTAL WEBPAGE

We strongly encourage the readers to refer to our supplemental webpage for more novel-view synthesis, static-dynamic disentanglement, localization, and scene editing results, as well as video comparisons with the state of the art.

Input t=0             Predicted intermediate frames             Input t=1

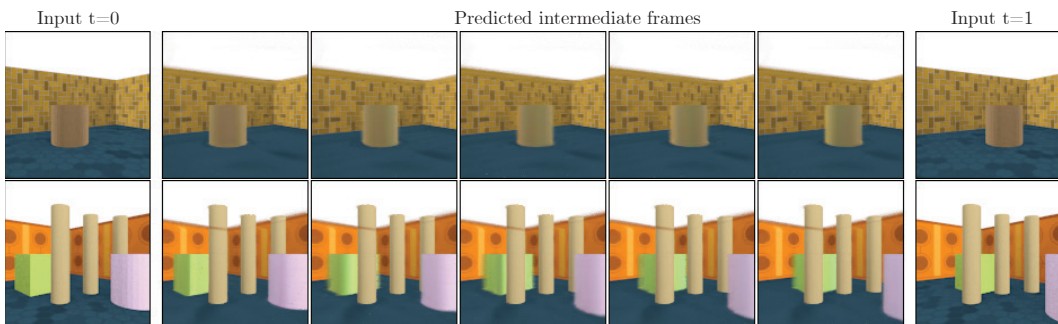

Figure 12: **Temporal 3D scene interpolation for novel view synthesis in space and time.**

