# OpenReview forum: "Neural Groundplans: Persistent Neural Scene Representations from a Single Image"
_ICLR.cc/2023/Conference — ICLR 2023 poster_

### Official Review · Reviewer_5tSi · 2022-10-23

**Confidence:** 3
**Clarity, Quality, Novelty And Reproducibility:** 1. The overall clarity and quality of…
**Correctness:** 4
**Technical Novelty And Significance:** 4
**Empirical Novelty And Significance:** 4
**Recommendation:** 6

**Strength And Weaknesses:**

Strength:

1. The authors tackle a challenging yet valuable problem, disentanglement of the movable and immovable components of the scene from a single image. They propose a neural scene representation called conditional neural groundplans and a self-supervised framework to disentangle static background and movable foreground objects given only a single image. The framework enables a variety of downstream tasks, such as extraction of object-centric 3D representations, novel view synthesis, instance-level segmentation, 3D bounding box prediction, and scene editing. The direction would interest a wide range of communities, including 3D scene understanding, 3D neural representation, disentangled representation learning, and novel view synthesis.
2. The proposed 3D persistent scene representation learning framework elegantly leverages ideas from BEV representation, neural representation learning, volume rendering, and compositing operation (Yuan et al., 2021). Moreover, the proposed method utilizes multi-view videos (in a self-supervised manner) at training time to learn to separately reconstruct static and movable components of the scene from a single image at test time. The authors present a self-supervised neural scene representation framework for general scene understanding tasks. The reviewer found the work valuable to the community.
3. The proposed 3D scene representations enable novel view synthesis and disentangle movable and immovable components of the scene via diverse qualitative evaluation on two datasets, CLEVR and CoSY. In particular, the qualitative results shown on CoSY are impressive, as shown in the manuscript and supplementary website.

Weaknesses:

1. Persistent scene representation: By sufficient qualitative analysis, the proposed 3D scene representations enable novel view synthesis and disentangle movable and immovable elements for different scene understanding tasks. One failure case shown on the top row of p.17 is that the framework cannot effectively synthesize the viewing perspective behind the box with a green door. The framework seems unable to capture the background correctly, yielding incorrect synthesis. Moreover, the box's location seems incorrect from the images on the left. The reviewer wonder if the performance can be improved with additional observations.
2. The authors propose to have two separate groundplans neural representations for static and dynamic, respectively. As we know from the literature (e.g., Yuan et al., 2021), it is possible to keep separate 3D neural representations for novel view synthesis. It seems reasonable to tackle the goal of this paper, i.e., achieve decent performance on a variety of tasks from a single image, with separate 3D neural representations based on the proposed framework. It would be great for the authors to elaborate on the technical challenge of this direction. That could motivate the community to tackle the direction collectively.
3. Differentiable rendering based on neural groundplans feature: the reviewer found it less intuitive to understand why the differentiable rendering would work using the groundplan features g_x. Specifically, g_x is obtained based on the weighted sum of the features along the y-axis. The groundplan feature g_x contains information in a particular pillar. Why is it a better feature for volume rendering? Empirically, the design choice renders better quality than STOA methods (i.e., PixelNeRF and uORF). Could the authors please elaborate more on this design choice?
4. The design choice of BEV representation: The authors utilize a particular operation to form BEV representations, i.e., the weighted sum of the features. However, there are other approaches, e.g., Roddick and Cipolla, 2020, Philion and Fidler, 2020 and Saha et al., 2022. It would be valuable for the audience to learn the difference between these operations for representation learning.
5. The reviewer is interested in the qualitative performance of novel view synthesis without using the contracted coordinate. With the analysis, readers would be able to visualize the difference.

**Summary Of The Paper:**

The authors proposed a method that maps a single 2D image of a scene to a persistent 3D scene representation, enabling novel view synthesis and disentangled representation of the movable and immovable components of the scene. The enabler is a new representation called conditional neural groudplans, motivated by the Birds’-eye-view representation. Because of the ability to separate the movable and immovable components of the scene, the representation enables a variety of downstream tasks, such as extraction of object-centric 3D representations, novel view synthesis, instance-level segmentation, 3D bounding box prediction, and scene editing. The authors conduct experiments on CLEVR and CoSY. The proposed method achieves state-of-the-art novel view synthesis compared with PixelNeRF and uORF on CLEVR. In addition, the proposed method obtain favorable instance segmentation performance compared with uORF on the CLEVR dataset.

**Summary Of The Review:**

The work presents a novel 3D persistent scene representation framework and demonstrates impressive novel view synthesis performance and scene understanding tasks such as instance segmentation and localization. The reviewer found the work novel and has the potential to stimulate the community to study the direction collectively. However, there are several concerns mentioned in the Weaknesses section. The reviewer would like to get feedbacks from the authors.

---

> ### Author Response · Authors · 2022-11-13
> **Author Response to Reviewer 5tSi**
>
> We thank the reviewer for their constructive feedback and suggestions.
>
> **Q: Inference of the background is incorrect as in the case of P17 top row. Would additional observations help?**
>
> **A:** Yes, additional observations would help in this case, in which the depth of the background was predicted incorrectly. An additional view that would observe these 3D points from a different angle could address this. We agree with your observation that in this example, the depth encoder encoded the green bus stop at incorrect depth. At this time, we could not track down this specific example, but more observations from different views would yield an improved reconstruction. Alternatively, we found an example where the model misappropriated the geometry when a single context view was provided [[LINK]](https://imgur.com/a/RElNKqh). In this example, we show the output renderings with additional observations along the rows.
>
> We observe that with a single input context image, the rear wheel in the output rendering is produced even though the rear wheel should not even be present in the output image frame. As more frames are provided, the output improves as the feature associated with the rear wheel is encoded at the correct depth. Please refer to Fig. 11 for results that show that more views help with uncertainty.
>
> **Q: Can you elaborate the technical challenges of this direction?**
>
> **A:** Thanks for the insightful suggestion. Inferring a persistent 3D representation based on the input 2D image enables prior based completion of the objects in the scene. Prior methods have focused on fitting on a single video sequence which doesn’t allow for learning of priors applicable across scenes. Two main challenges are 1) reconstructing the 3D scene from a single or sparse image observations, which requires learning a strong prior over 3D geometry and appearance, and 2) learning a prior over which objects are likely to move or not from observing multi-view video at training time. At test time, we can use this prior to disentangle static from movable objects from a single image, thus, without observing any object motion. This 3D representation learning from images involves learning to lift the features to 3D space. And the 3D prior is learned over the 3D representation by observing objects and scenes from different views.
>
> **Q: Differentiable rendering based on neural groundplans feature: the reviewer found it less intuitive to understand why the differentiable rendering would work using the groundplan features g_x. Specifically, g_x is obtained based on the weighted sum of the features along the y-axis. The groundplan feature g_x contains information in a particular pillar. Why is it a better feature for volume rendering? Empirically, the design choice renders better quality than STOA methods (i.e., PixelNeRF and uORF). Could the authors please elaborate more on this design choice?**
>
> **A:** The neural rendering works using the feature g_x where g_x is a latent for a given (x, z) location on the ground plane. This latent holds the color and density information for all points along the height (y-axis) at that (x, z) location. The improved quality is a result of having a persistent 3D representation, neural groundplans, which enables CNNs to complete and refine the 3D objects as they operate on a 2D grid of features without issues related to perspective distortion. This is not the case with existing methods which encoder features only in the image plane (PixelNeRF, uORF).
>
> **Q: The reviewer is interested in the qualitative performance of novel view synthesis without using the contracted coordinate. With the analysis, readers would be able to visualize the difference.**
>
> **A:** We are running the experiment and will post the results in the upcoming days once the training completes.

---

> > ### Author Response · Authors · 2022-11-18
> > **Qualitative performance of novel view synthesis without using contracted coordinates**
> >
> > We ran the experiment without the use of contracted coordinates.
> >
> > Here are the results: [[LINK]](https://imgur.com/a/9gPpnK3)
> >
> > In this setting, the visual quality suffers for all objects, as the groundplan needs to linearly explain a larger area, resulting in a grid with lower effective resolution. This low resolution leads to fewer set of latents to express the objects, impacting the visual quality of the renderings.

---

> ### Author Response · Authors · 2022-11-30
> **Looking forward to your reply**
>
> Dear Reviewer 5tSi,
>
> Thank you again for your time. Since the deadline for discussion is approaching, we hope to have further conversation with you to see if our response resolved your concerns. We would appreciate if you kindly checked our response. Please let us know if we can offer any further clarifications.
>
> Thanks,
>
> Authors

---

### Official Review · Reviewer_eeHh · 2022-10-23

**Confidence:** 4
**Correctness:** 3
**Technical Novelty And Significance:** 3
**Empirical Novelty And Significance:** 3
**Recommendation:** 6

**Clarity, Quality, Novelty And Reproducibility:**

Given the results, the reviewer feel that this work is in good quality. And based on my expertise, it presents good originality.

However, as stated in the weakness, the clarity of the paper needs further improvement. And although the authors give some details of the network in the appendix, the reviewer is not optimistic that it can be reproduced easily without code.

**Strength And Weaknesses:**

====> Strength
1. The authors introduced the bird’s eye-view (BEV) representation as a scene representation, which is novel in the area of neural rendering.
2. The disentanglement of the static and moving component of the scene adds more contribution to the proposed method. Although it is not new to automatically separate these components in the neural rendering community, using a vanilla 2D CNN to separate the feature representation is new to the reviewer.
3. The authors show that using a simple heuristic method, they can generate instance segmentation and bounding boxes for the dynamic components in the scene.
4. Good qualitative results are given.

====> Weakness
1. The details of the neural groundplans is confusing:

    i. In page 4, there are multiple $x$ such as $x'$, $x$, $x_c$, $x_i$. Their shape should be clarified. Whether they belong to 2D or 3D point, and also their coordinate system (i.e., the image frame coordinate or the world coordinate, and how to transform between these coordinates using the camera intrinsic)

    ii. For equation like $v(x ′ ) = F(π(C^{-1}(x ′ )))$, $f(x ′ ) = ...$, the parentheses here denote indexing the feature volume $v$ and feature tensor $F$. However, it is easy to confuse them with the functions like $\pi()$ and $D()$. The authors should consider using other formulation.

    iii.The $\pi(\cdot)$ is not clear. What is the meaning of it and how to learn $\pi(\cdot)$, is it learned per image I?

    iv. ``$g_x$ are the groundplan features for the point $x$ computed by projecting the coordinates onto the groundplane...'' How is the projection conducted? Is it projected by the camera extrinsic？

    v. In page 5, the authors say ``The per-frame static groundplans are pooled to obtain a single, time-independent static groundplan.'' How is the pooling process performed? The authors need to detail it.

2. The task setting is not clear. The authors should consider formulate the problem setting in a separated subsection (e.g., train one model for multiple scenes, all images are calibrated, the view density per scene, etc.)

3. It seems the compared baselines like pixelNerf and uORF are not proposed for the topic of separating dynamic and static representation. It would be fairer to compare with ``Neuraldiff: Segmenting 3d objects that move in egocentric videos.'' that used different components for the moving and static objects.

**Summary Of The Paper:**

This paper proposes a bird’s eye-view (BEV) representation for a 3D scene. The representation is disentangled into the static part and the dynamic part by a 2D CNN. The authors train their representation network on multiple dynamic videos with multiple views. When given a new image, the network can generate the implicit 3D representation and decompose the scene automatically. Differential rendering is used to generate novel view images. The authors conducted experiments on two datasets and showed impressive results.

**Summary Of The Review:**

Overall, the paper proposed a new representation for the neural rendering, and is novel to me. However, the clarity of the paper needs further improvement. The reviewer looks forward to the author feedback and will consider increase my rating if the discussion can address my questions.

=====
The authors addressed most of my concerns during the discussion and I would increase my rating.

---

> ### Author Response · Authors · 2022-11-13
> **Author Response to Reviewer eeHh**
>
> We thank the reviewer for the detailed and constructive review.
>
> **Q: Clarification on description of neural groundplans**
>
> **A:** Thanks for suggesting changes on the description of the method to improve on the clarity. We have made the following changes on page 4, marked in blue in the updated manuscript.
> - All points on page 4 are 3D. We have defined the shapes of all the points.
> - Changed the notation for indexing of the feature tensor using square brackets.
> - \pi() is an analytic projection function projecting a 3D point in world coordinates on the image plane using the camera intrinsics and extrinsics. It does not have any learnable parameters. We have added the details of the projection of a 3D point on the image plane and on the ground plane in the supplementary section A.1.
> - The per-frame static groundplans are mean-pooled to get a common groundplan across timesteps. We have updated this on page 5 in the updated manuscript.
>
> **Q: The task setting is not clear. The authors should consider formulate the problem setting in a separated subsection.**
>
> **A:** Thank you for your suggestion. We have succinctly formulated the problem setting at the beginning of section 3 in the revised paper.
>
> **Q: It seems the compared baselines like pixelNerf and uORF are not proposed for the topic of separating dynamic and static representation. It would be fairer to compare with ``Neuraldiff: Segmenting 3d objects that move in egocentric videos.'' that used different components for the moving and static objects.**
>
> **A:** While the paper proposed by the reviewer, NeuralDiff, and many other papers such as “D2NeRF: Self-Supervised Decoupling of Dynamic and Static Objects from a Monocular Video”, “Dynamic View Synthesis from Dynamic Monocular Video” perform static-dynamic disentanglement, these methods are all scene-specific, i.e., they require optimization for each scene separately, and require many frames of the scene captured over both time as well as diverse camera poses.
>
> This problem setting is entirely different from the problem setting we tackle: We tackle the problem of reconstructing a 3D scene from sparse image observations (as few as a single image) captured at a single timestep. At test time, our method reconstructs an unseen 3D scene from a single image, and separately reconstructs parts of the scene that are movable and parts of the scene that are static, based on prior learned by observing multi-view videos at training time. The methods the reviewer cites can simply not be applied to this problem setting to achieve the desired goal as they do not learn a prior that can be applied to a single image. In contrast, pixelNeRF and uORF both tackle the problem of 3D reconstruction from a single image, which is why we chose them as baselines for our approach.

---

> > ### Comment · Reviewer_eeHh · 2022-11-24
> > **Reviewer reply**
> >
> > Thanks for the clarification. The reviewer is satisfied with the updated version and will improve my rating.
> > An extra modification is needed for the future version: the $v(x′)$ should also be $v[x']$

---

> > > ### Author Response · Authors · 2022-11-30
> > > **Author response**
> > >
> > > Thank you for your time and suggestions to improve the paper. We will make the suggested change in the notation in the future version.

---

### Official Review · Reviewer_U5HL · 2022-10-24

**Confidence:** 4
**Correctness:** 4
**Technical Novelty And Significance:** 3
**Empirical Novelty And Significance:** 3
**Recommendation:** 6

**Clarity, Quality, Novelty And Reproducibility:**

Quality is overall high, with thorough evaluations and a convincing and coherent story. Clarity-wise, I find the paper easy to follow and pleasant to read. There are a few of details that I think the authors should provide more clarification (last two points in weaknesses).

I think all major components of the work have been seen in some previous work. However, they are nicely put together to achieve a 3D presentation with the combination of some unique strengths (single-image 3D scene inference, movable object editing, self-supervised training).

There are a decent amount of details in the paper, and the authors promise code and data release, so I think reproducibility is not a concern.

**Strength And Weaknesses:**

Strengths include:
* The self-supervised training removes the need for GT annotation, e.g., object bounding boxes, for the supported downstream tasks.
* The decomposition of static and movable object feature spaces looks like quite an elegant solution to utilize the inductive bias in multi-view videos while enabling capabilities such as segmentation and object-level scene editing.
* The results overall look promising. Single-view/few-view novel view synthesis is a very challenging task, and the comparisons with PixelNeRF and uORF clearly demonstrated the advantages in synthesis quality. The analysis of adding more views (Fig.10) is strong evidence that the learned 3D representation can figure out how 2D views are related to, and hence help improve, 3D reconstruction.

Weaknesses include:
* There is no study of the generalization capability of the model. It seems likely that the learned models can overfit the limited appearances in the dataset (a few types of shapes/objects with mostly uniform colors).
* Related to the last point, the evaluation is completely done on synthesized datasets, and it's not clear how well the model can learn real-world data with much more diverse appearances. Since the method does not require GT annotations, some evaluation on real-world data does not seem too difficult.
* The are some quality issues with the presented view synthesis results. For example, while unobserved regions are understandably blurry, why are observed regions also not sharp? We can see from Fig.10 that adding more views does not help the quality of observed regions any further.
* Training requires multi-view videos. I can't find clarification on whether these need to be from static cameras. Nothing in the model appears to have such a requirement, but if so, that's a strong limitation. Moreover, why can't the model be trained on frames from a single moving camera?
* It's not clearly described how scene editing is done. It says that's enabled by "directly editing the neural groundplan". Does that mean editing the (movable object) feature space? The feature space is not sparse (i.e. locations without objects have non-zero features), so how does cropping in such space work, and why can that make sense?

**Summary Of The Paper:**

The paper describes a 3D scene representation built from 2D observations.

The representation allows the decomposition of static and movable objects in the scene, and can support tasks including novel view synthesis, 3D instance segmentation, and object-level scene editing. The authors propose a self-supervised training method for learning such a representation.

Evaluations are conducted on two synthetic datasets.

**Summary Of The Review:**

The major highlights of the paper are the use of multi-view videos as self-supervision and, consequently, the ability to decompose static/movable objects to support segmentation and editing. The results are promising, although they appear to be somewhat preliminary.

My primary concerns are around generalization and the lack of evidence on how the approach performs on real, non-synthetic images. There are also some details requiring clarifications that should be easy fixes.

---

> ### Author Response · Authors · 2022-11-13
> **Author Response to Reviewer U5HL (1/2)**
>
> We thank the reviewer for their constructive suggestions.
>
> **Q: The evaluation is completely done on synthesized datasets, and it's not clear how well the model can learn real-world data with much more diverse appearances.**
>
> **A:** Current real datasets such as KITTI 360 and Argoverse do not have camera setups where the 3D points in the scene are observed by multiple cameras with wide baselines. Moreover, we are not aware of any existing method that learns priors for single image-based full 3D reconstruction by training on these datasets. These datasets unfortunately do not provide enough supervision for learning a full 3D reconstruction of the scene at any one timestep.
>
> Our fundamental contribution is demonstrating, for the first time, that it is feasible to learn to reconstruct a 3D scene representation from a single image that is aware of static and movable parts of the 3D scene. This is different from recent photorealistic results demonstrated by fitting on a single scene, as our method needs to learn a prior over 3D appearance and what object are static and movable to be able to apply it on a single image. Let us explain in detail:
>
> Recent results have achieved photo-realistic novel view synthesis on real-world scenes via fitting a single scene given many observations of that scene. In this regime, static-dynamic disentanglement has also been demonstrated, in methods such as STaR, NeuralDiff, or D2NeRF. We note that our proposed scene representation can be similarly overfit for novel view synthesis of a real-world scene, as demonstrated in this video of a RealEstate10k scene  [[LINK]](https://streamable.com/z4jxu1).
>
> However, in our research we attempt to infer a conditional neural groundplan, a 3D scene representation, which performs static-dynamic disentanglement of 3D scenes from a **single or very few** image observations. Our method is trained self-supervised on RGB inputs with corresponding camera poses from wide baselines, and requires no other form of annotations. This data is employed to learn a prior which is employed to plausibly complete the 3D appearance and differentiate between the static and movable objects. This is a **significantly** more challenging problem as it requires learning prior over 3D appearance of objects. **There does not currently exist an approach that can reconstruct a full 3D NeRF (rendering views from wide baselines) of a real-world scene from a single image on real datasets.** The only work towards this is the reconstruction of extremely small-baseline, 2.5 D multi-plane images from a single image, which is far from reconstructing a full 3D representation.
>
> We benchmark our approach with a strong baseline that makes identical assumptions, which is pixelNeRF, and decidedly outperform it. Our CoSY scenes are significantly more complex than Shapenet results that were previously demonstrated in the single-image regime. Note that uORF, a very recent 2021 paper, demonstrates results only on CLEVR and simple, single-room-scale scenes!
>
> We would thus like to assert that we already demonstrate a significant step in terms of capabilities with the current, synthetic, CoSY and CLEVR dataset. While some camera setups, such as traffic cameras, or sports footage, would be suitable, there does not exist any large datasets in these settings. Further progress in algorithms that can reason about the dynamic 3D structure from small baseline training data is a path towards training on a wide range of real world datasets, but out of scope of our current work. We take the first step towards solving this fundamental problem, and believe that future work will address many of the current limitations.

---

> > ### Author Response · Authors · 2022-11-13
> > **Author Response to Reviewer U5HL (2/2)**
> >
> > **Q: There are some quality issues with the presented view synthesis results. For example, while unobserved regions are understandably blurry, why are observed regions also not sharp? We can see from Fig. 10 that adding more views does not help the quality of observed regions any further.**
> >
> > **A:** The sharpness of the observed regions is limited by two key factors.
> > First, we are in the regime of prior-based reconstruction from a few images. This is in contrast to the regime of single-scene overfitting (e.g. NeRF)  in which prior work has demonstrated photo-realistic results. In our task setup of prior based reconstruction, a certain degree of uncertainty exists. For instance, there is uncertainty about the exact depth and geometry of the 3D scene given the input images. In these cases, the model will learn to blur proportionate to the amount of uncertainty. We note that this is not a limitation of our method specifically - all prior-based 3D reconstruction methods share this property. We outperform pixelNeRF, a strong baseline in this regime of 3D reconstruction from few images, both quantitatively and qualitatively.
> >
> > Second, our rendering quality is limited by computational cost. In contrast to single-scene methods, our method needs to fit not only the differentiable rendering in GPU memory, but also the whole inference pipeline - convnets, 3D lifting, groundplans, etc. This limits the resolution of the groundplans (a car is expressed by ~6 latents) as well as the number of volume rendering samples. Increasing computational budget would lead to better renderings, but is outside our compute budget.
> >
> > Fig. 10 shows that the model resolves the uncertainty due to unobserved 3D regions as more views are provided as input views. The region observed in input view 1 does not change significantly as the rendering quality is limited because of the reason above. We have added this discussion to the supplementary section B.2 along with Figure 10.
> >
> > **Q: Requirement of static camera?**
> >
> > **A:** The training regimen and construction of the model allow for training using multi-view videos captured with moving cameras.
> >
> > **Q: It's not clearly described how scene editing is done.**
> >
> > **A:** We have added a discussion of how scene editing is performed to the supplementary section A.5, which we elaborate here.
> >
> > Scene editing is performed by editing the dynamic groundplan. We may, for instance, copy the latents of a car, and paste them at another location in the grid, using bilinear interpolation if grid elements do not line up perfectly. Similarly, we may remove a car by setting its latents to surrounding latents that encode zero density. We may similarly translate and rotate latents to different locations, and may even copy a car from one scene to the dynamic groundplan of a different scene.

---

> > > ### Comment · Reviewer_U5HL · 2022-11-30
> > > **Re: Author Response to Reviewer U5HL**
> > >
> > > I appreciate the detailed responses.
> > >
> > > Q1 & Q2 -- the explanations are indeed reasonable. However, the raised issues (unproven generalizability and blurry reconstruction) are still limitations of the approach and/or weaknesses of the submission nonetheless.
> > >
> > > Q3 -- this could be a point worth some clarification in the paper.
> > >
> > > Q4 -- thanks for the clarification and the addition to the supplementary.

---

> ### Author Response · Authors · 2022-11-30
> **Looking forward to your reply**
>
> Dear Reviewer U5HL,
>
> Thank you again for your time. Since the deadline for discussion is approaching, we hope to have further conversation with you to see if our response resolved your concerns. We would appreciate if you kindly checked our response. Please let us know if we can offer any further clarifications.
>
> Thanks,
>
> Authors

---

### Official Review · Reviewer_KVRy · 2022-10-24

**Confidence:** 3
**Correctness:** 3
**Technical Novelty And Significance:** 3
**Empirical Novelty And Significance:** Not applicable
**Recommendation:** 6

**Clarity, Quality, Novelty And Reproducibility:**

The paper is well-organized and clearly written. Ideas are novel. Results in the paper are easily reproducible.


**Strength And Weaknesses:**

Strength：
1. The conditional neural groundplans proposed are intriguing. This allows efficient processing of scene appearance and geometry directly in 3D, through the self-supervised learning of conditional neural groundplans, a hybrid discrete continuous representation of 3D neural scenes.
2. The proposed disentanglement of static background and movable foreground objects is novel. This approach uses object motion for training an encoder-based system that can reconstruct scenes from a single image instead of decomposing video frames based on scene specifics.
3. The proposed method is promising for understanding 3D from a single image. The dynamic groundplan encodes 3D geometry, allowing us to segment and animate objects in 3D using single images and 3D bounding boxes.

Weaknesses：
1. The biggest weakness is that only results on simple synthetic data are presented. Currently, there are a number of well-annotated 3D driving scenes available. An example is KITTI 360 (https://www.cvlibs.net/datasets/kitti-360/). Could the authors explain why results on the real datasets are not presented? Is it because of some specific annotation (such as BEV correspondence or the number of views) needed by the method? Otherwise, the generalizability on natural scenes and the practical value of the method are questioned.
2. I think there are some ambiguity behind the proposed Static-Dynamic Disentanglement. A definition of Static and Dynamic should be made clear in the proposed framework. A car remaining still in the sequence will be considered as Static. Considering a situation when two moving and static cars looks similar or exactly the same, how can we distinguish from a single image if a car is static or dynamic? The method should mention the above cases as an assumption, or making a clear definition of being Static and Dynamic.
3. Groundplans implicitly assume that objects are aligned on a plane. However, in real life, it is common that scenes/objects are not placed on a flat ground plane (e.g., a curved plane). The ability of persistent 3D scene representation is a little overclaimed, if we consider the implicit assumption behind groundplans.


**Summary Of The Paper:**

An approach is presented in this paper for mapping 2D image observations to persistent 3D scene representations in order to create a novel view synthesis and disentangle the movable and immovable components of the scene. A ground-aligned 2D feature grid based on conditional neural groundplans can be employed as a memory-efficient representation of scene data. This notion is inspired by the bird's-eye-view (BEV) representation commonly used in vision and robotics. Self-supervised learning is conducted using differentiable rendering from unlabeled multi-view observations, and the method learns to complete the geometry and appearance of occluded regions.


**Summary Of The Review:**

The method is good and the idea is interesting. But only synthetic data is used for evaluation, and the access of real data should not be a problem.

---

> ### Author Response · Authors · 2022-11-12
> **Author Response to Reviewer KVRy**
>
> We thank the reviewer for their thoughtful comments.
>
> **Q: Could the authors explain why results on the real datasets are not presented? Is it because of some specific annotation (such as BEV correspondence or the number of views) needed by the method?**
>
> **A:** Current real datasets such as KITTI 360 and Argoverse do not have camera setups where the 3D points in the scene are observed by multiple cameras with wide baselines. Moreover, we are not aware of any existing method that learns priors for single image-based full 3D reconstruction by training on these datasets. These datasets unfortunately do not provide enough supervision for learning a full 3D reconstruction of the scene at any one timestep.
>
> Our fundamental contribution is demonstrating, for the first time, that it is feasible to learn to reconstruct a 3D scene representation from a single image employing a prior on the 3D appearance of objects, and static and movable parts of the 3D scene. This is different from the recent works on fitting a single scene given many observations of that scene. In this regime, static-dynamic disentanglement has also been demonstrated by methods such as STaR, NeuralDiff, or D2NeRF. Note that our proposed scene representation can be similarly fit for novel view synthesis of a single real-world scene, as demonstrated in this video of a RealEstate10k scene [[LINK]](https://streamable.com/z4jxu1).
>
> In our research we attempt to infer a conditional neural groundplan, a 3D scene representation, which performs static-dynamic disentanglement of 3D scenes from a single or very few image observations. Our method is trained self-supervised on RGB inputs with corresponding camera poses from wide baselines, and requires no other form of annotations. This data is employed to learn a prior which is employed to plausibly complete the 3D appearance and differentiate between the static and movable objects.
>
> We would thus like to assert that we already demonstrate a significant step in terms of capabilities with the current, synthetic, CoSY and CLEVR dataset. While some camera setups, such as traffic cameras, or sports footage, would be suitable, there does not exist any large datasets in these settings. Further progress in algorithms that can reason about the dynamic 3D structure from small baseline training data is a path towards training on a wide range of real world datasets, but out of scope of our current work. We take the first step towards solving this fundamental problem, and believe that future work will address many of the current limitations.
>
> **Q: A car remaining still in the sequence will be considered as Static. Considering a situation when two moving and static cars looks similar or exactly the same, how can we distinguish from a single image if a car is static or dynamic? The method should mention the above cases as an assumption, or making a clear definition of being Static and Dynamic.**
>
> **A:** We agree that the word “dynamic” is overloaded. With “dynamic”, we mean “movable” - i.e., given observations in the training set, one would expect the object to be movable, not that it is necessarily currently moving. For instance, at training time, we regularly see cars move - therefore, at test time, given a single image, we want to interpret that the car as movable. In the scenario that the reviewer raises, our model would learn to flag both cars as “movable”.
>
> We have clarified the use of “dynamic” in the paper and formally defined “static” and “dynamic” in the introduction - please refer to the paper revision.
>
> **Q: Groundplans implicitly assume that objects are aligned on a plane. However, in real life, it is common that scenes/objects are not placed on a flat ground plane (e.g., a curved plane).**
> **A:** The design choice of groundplans does not assume flat ground planes, and can reconstruct curved planes just in the same way as it can encode any other type of 3D geometry in the scene such as building or cars. Specifically, a latent at an (x, z) location on the groundplan encodes the information about if *any* points along the height axis might be occupied and what the color of the points. The only downside to curved ground planes is a reduction in the effective resolution of our scene representation. For instance, if the street has a 45 degree incline, a grid element with sidelength 1 now has to encode information about a street segment of sidelength $1/cos(45 \degree) = 1.41$.

---

> ### Author Response · Authors · 2022-11-30
> **Looking forward to your reply**
>
> Dear Reviewer KVRy,
>
> Thank you again for your time. Since the deadline for discussion is approaching, we hope to have further conversation with you to see if our response resolved your concerns. We would appreciate if you kindly checked our response. Please let us know if we can offer any further clarifications.
>
> Thanks,
>
> Authors

---

### Author Response · Authors · 2022-11-13
**Author Response**

We thank the reviewers for their careful reading, and detailed and considerate feedback.

We are glad that reviewers agree that conditional neural ground plans are an effective representation, ("neural groundplans proposed are intriguing.", “promising for understanding 3D from a single image” (KvRY), “results overall look promising” (U5HL), “Good qualitative results are given” (eeHh), “demonstrates impressive novel view synthesis performance and scene understanding tasks” (5tSi)). The reviewers have requested for several clarifications. Below, we respond to the questions raised by the reviewers and discuss how we integrate the feedback in the paper. The paper is revised with blue colored text to easily see the revisions.

---

### Decision · Program_Chairs · 2023-01-20

**Decision:**

Accept: poster

**Justification For Why Not Higher Score:**

The article had negative scores. The reviewers pointed out several weaknesses and the authors addressed many of them. However, there are some points that require further research. Overall the contribution is good.

**Justification For Why Not Lower Score:**

The article is worth publishing since it has important contributions even there is need of more research.

**Metareview: Summary, Strengths And Weaknesses:**

The authors propose neural ground plans 3D representations to represent 3D extracted from a single image. The method obtains good qualitative results showing how to render new views for scene understanding. The reviewers had initially several concerns like:
- Results only on synthetic datasets
- Difficulties to determine if a vehicle is moving or static
- Some baseline comparisons


**Note From Pc:**

if the above contains the word "oral" or "spotlight" please see: "oral" presentation means -> notable-top-5% and "spotlight" means -> notable-top-25%. As stated in our emails, we are disassociating presentation type from AC recommendations